# Interaction between Kidney-Bean Polysaccharides and Duck Myofibrillar Protein as Affected by Ultrasonication: Effects on Gel Properties and Structure

**DOI:** 10.3390/foods11243998

**Published:** 2022-12-09

**Authors:** Yang Wu, Qiwei Du, Xiankang Fan, Changyu Zhou, Jun He, Yangying Sun, Qiang Xia, Daodong Pan

**Affiliations:** 1State Key Laboratory for Managing Biotic and Chemical Threats to the Quality and Safety of Agro-Products, Ningbo University, Ningbo 315048, China; 2Key Laboratory of Animal Protein Food Processing Technology of Zhejiang Province, College of Food and Pharmaceutical Science, Ningbo University, Ningbo 315211, China

**Keywords:** ultrasound, dietary fiber, myofibrillar protein, gel properties, structure

## Abstract

The interaction of polysaccharides–protein with varied origins and structures provides opportunities for tailoring the physicochemical qualities of food protein-based materials. This work examined the feasibility of ultrasound-modified interaction between kidney bean dietary fiber (KSDF) and duck myofibrillar proteins (MP) to improve the physicochemical properties of the gel matrices. Accordingly, gel strength, water holding capacity, solubility, chemical interaction, secondary structure, and network structure of MP were determined. The addition of KSDF combined with the ultrasound treatment contributed to the improved water retention capability, G’ values, and the reduced particle size of protein molecules, corresponding with the formation of dense pore-like structures. The results demonstrated that 1% KSDF and ultrasonication at 400 W significantly enhanced gel strength by up to 109.58% and the solubility increased by 213.42%. The proportion of α-helices of MP gels treated with 1% KSDF and ultrasonication at 400 W was significantly increased. The sonication-mediated KSDF–MP interaction significantly improved hydrophobic interactions of the proteins, thus explaining the denser network structure of the MP gels incorporated KSDF with ultrasound treatments. These results demonstrated the role of ultrasonication treatments in modifying KSDF–protein interaction to improve the gel and structural properties of the MP gels.

## 1. Introduction

Asia accounts for approximately 84.2% of the world’s duck meat production, among which China has always been one of the leading producers of duck meat. However, traditional duck meat products are developing slowly, and various forms of minced duck meat products to stimulate consumer demand are needed [1]. There is a tendency toward creating various functional low-temperature minced meat products due to the gradual rise in the number of sub-healthy people [2]. However, the loss of water and oil in traditional low-temperature minced meat products has a serious impact on the sensory quality of meat products. Additionally, the low gel qualities of duck breast significantly lower the sensory quality of meat products [3]. Therefore, the addition of starch, polysaccharides, and dietary fiber to meat products has been investigated to improve the gelation properties of myofibrillar proteins (MP) for improving the quality of meat products.

Soluble dietary fiber (SDF) dissolved in water can absorb water, swell, and can be fermented by microorganisms in the large intestine. Although kidney beans are widely grown in China, they are mostly neglected as a source of dietary fiber. Kidney beans as an excellent natural source of dietary fiber play an essential role in reducing body weight, balancing blood sugar levels, reducing the risk of heart disease, and improving intestinal health [4]. Kidney beans’ soluble dietary fiber (KSDF) not only plays a significant physiological role, but it is also frequently employed in food processing and to enhance gel characteristics [5]. Some surveys have shown that the addition of oat-insoluble dietary fiber to meat products significantly enhanced the gel strength of meat products, and also improved the color and texture [6]. Existing studies have also found that the addition of sugarcane dietary fiber can effectively reduce the water channels in the gel network and make the gel network structure dense [7]. Distinct dietary fibers typically have different gel properties, and the extent that a single dietary fiber may enhance the gel properties of MP is frequently constrained. The majority of current research is restricted to looking at how a single dietary fiber affects the macroscopic characteristics of meat products [8], which restricts the use of dietary fiber in food processing.

As a new non-thermal processing technology, ultrasound, which is a mechanical vibration wave over 20 kHz, is widely used in the chemical, biological, medicinal, and food industries [9]. It has been shown that ultrasound can modify the structural and functional properties of proteins, such as improving solubility, and hydrophobic emulsion gelation [10]. In our previous study, the effect of different ultrasound powers on the structure of chicken breast protein was evaluated, revealing that proper sonication can effectively preserve the thermal stability of the chicken breast protein and reduce the loss of energy storage modulus [11]. On the basis of this theory, this study applied ultrasound to a composite system of dietary fiber and protein with a view to further improving the gelation properties of the protein. Additionally, the present research has only examined how ultrasound can enhance gel characteristics; it has not looked at how ultrasound affects interactions between proteins and dietary fiber.

Therefore, this study aimed to explore the effect of ultrasound on the interaction between dietary fiber and duck myogenic fibronectin and to investigate the link between the physicochemical properties of KSDF–MP and the properties of gels. The results of this study could provide some theoretical basis for the development of minced duck products.

## 2. Materials and Methods

### 2.1. Materials and Reagents

The duck breast meat was purchased from Huaying poultry group (Xinyang, China), and kidney bean soluble dietary fiber was provided by Dongfeng Biotechnology (Xi’an, China). All chemicals of analytical grade used in the work were from commercial sources.

### 2.2. Extraction of Myofibrillar Proteins

The extraction method of MP was referred to as the method of Zhuang et al. [12]. The ground duck breast was added to 4 vol of Tris-EDTA (0.1 mol/L Tris, 10 mmol/L EDTA) extract and stirred, and homogenized in an ice bath at 10,000 r/min for 60 s. After homogenization, the supernatant was removed by centrifugation at 6000 r/min for 20 min, and then 4 vol of SSS reagent (containing 0.1 mol/L KCl, 0.02 mol/L K_2_HPO_4_/KH_2_PO_4_, 2 mmol/L MgCl_2_, 1 mmol/L EGTA, pH 7.0) was added, shaken well, and centrifuged at 8000 r/min for 10 min. The supernatant was removed, with the precipitate as the purified MP.

### 2.3. Sample Treatment

The extracted myofibrillar protein was diluted to 40 mg/mL with 0.02 mol/L PBS buffer, 1% of KSDF was added and mixed, then the samples were divided into 10 ml beakers for sonication (0 W, 200 W, 400 W, and 600 W). The ultrasound procedure lasts 5 min with an intermittent mode (3 s on and 6 s off). Additionally, the control group (CK) was the sample without adding dietary fiber or ultrasound treatment. The samples were refrigerated at 4 °C. The treated samples of each group were heated in a water bath from 25 to 85 °C for 30 min, then cooled in an ice bath and placed in a 4 °C refrigerator overnight.

### 2.4. Gel Properties

#### 2.4.1. Gel Strength

The gel strength was determined according to the method of Li et al. [13] with few modifications. Testing was performed using the P0.5 probe of the physical property analyzer (TA.XT. PlusC, Stable Micro Systems Ltd., Godalming, UK) with a pre-test rate and mid-test rate of 1 mm/s, a post-test rate of 5 mm/s, a deformation rate of 40%, and a trigger force of 3 g.

#### 2.4.2. Water Holding Capacity

The determination of water holding capacity (WHC) is based on the method of Jiang et al. [14]. By centrifugation, a certain mass of protein gel was put into a centrifuge tube and centrifuged at 10,000 r/min for 5 min, and the mass before and after centrifugation was recorded, respectively, and the WHC was defined as the following equation:(1)$$WHC\left(\%\right)=\left(M_{2}-M_{3}\right)/\left(M_{1}-M_{3}\right)\times 100$$
where *M*_1_ indicates the total mass before centrifugation, *M*_2_ indicates the total mass after centrifugation and *M*_3_ indicates the mass of the centrifuge tube.

#### 2.4.3. Dynamic Rheological Testing

The rheological properties of the sol sample were analyzed using a DHR-2 rheometer equipped with a 40 mm probe [15], and the sol was oscillated from 20 to 85 °C at 2 °C/min with a linear ramping frequency of 0.1 Hz. The storage modulus (G’) was recorded.

#### 2.4.4. Low-Field Nuclear Magnetic Resonance (NMR)

Relaxation time (T2) and water distribution were measured using a Niumag-pulsed NMR analyzer (MicroMR12-025V, Niumag Analytical Instruments Corporation, Suzhou, China) by the Carr-Puecell-Meiboom-Gill (CPMP), which equated to a proton resonance frequency of 12 MHz. The samples (weight of 2.5 ± 0.1 g) were placed into a cylindrical glass tube (diameter 22 mm, height 50 mm) at room temperature. The relaxation data were analyzed by the method of Guo et al. [16] using Muti-Exp Inv Analysis software (Niumag Electric Corp, Suzhou, China). The proportion of the water distribution of each component was calculated by normalizing the peak area of the curve.

#### 2.4.5. Scanning Electron Microscopy

Protein gel samples were sliced into 2 mm^3^ cubes, fixed in liquid nitrogen, and then lyophilized and the microstructure was observed at 10.0 kV using S3400N (Hitachi, Tokyo, Japan) at 100× magnification [17].

### 2.5. Physicochemical Properties

#### 2.5.1. Solubility

The solubility was determined with reference to Pan et al. [18]. The protein solution sample (5 mg/mL) was left for 1 hour at 4 °C and centrifuged at 9000 r/min for 15 min. The protein concentration of the supernatant was measured, and the solubility was expressed as a ratio of the supernatant protein concentration to the stock solution protein concentration.

#### 2.5.2. Surface Hydrophobicity

According to the bromophenol blue (BPB)-binding method of Pan et al. [18]. The protein concentration was adjusted to 5 mg/mL. An amount of 1 mL of the sample was added to 200 μL of 1% bromophenol blue and mixed for 10 min, centrifuged at 9000 r/min for 10 min. The supernatant was diluted 10 times and the absorbance value was measured at 595 nm. Surface hydrophobicity was expressed as below:(2)$$BPB\;bound\left(\mu g\right)=\left(A_{con}-A_{sample}\right)/A_{con}\times 200$$
where *A_con_* indicates the absorbance value of the phosphate buffer without MP addition, and *A_sample_* indicates the absorbance value of the supernatant of the sample.

#### 2.5.3. Total SH

The total sulfhydryl group was determined by reference to Alavi et al. [19]. An amount of 5 mg/mL of MP was diluted 10 times with Tris-glycine buffer containing 10 mM of DTNB and the absorbance at 412 nm was recorded at room temperature for 30 min. A molar extinction coefficient of 13,600 L M^−1^ cm^−1^ was used to calculate the SH content.

#### 2.5.4. SDS-PAGE

SDS-PAGE was performed according to the published method [20]. The sample protein concentration was adjusted to 1 mg/mL, mixed with the loading buffer, and set aside in a metal bath at 100 °C for 10 min. SDS-PAGE was performed using spectral standard protein labeling, 80 V for 30 min and 120 V for 90 min.

#### 2.5.5. Particle Size and Zeta Potential

The micron particle size was determined using a micron laser particle sizer (HELOS-OASIS) and the zeta potential was determined using a Zetasizer **Nano ZS** (Malvern Instruments Ltd., Malvern, UK), tuning the protein to the same concentration to reduce multiple scattering errors [21].

#### 2.5.6. Fourier Transform Infrared Spectroscopy (FTIR)

FTIR was performed based on the previous method [22]. After freeze-drying, the proteolytic samples were mixed and ground with potassium bromide powder at a mass ratio of 1:25, and FTIR spectra were scanned in the range of 400~4000 cm^−1^.

#### 2.5.7. Chemical Forces

The chemical forces of the MP gels were determined with reference to Shi et al. [23]. 0.5 g of gel was weighed and homogenized with 4.5 mL of S1 (0.05 M NaCl), S2 (0.6 M NaCl), S3 (0.6 M NaCl, 1.5 M urea), S4 (0.6 M NaCl, 8 M urea), S5 (0.6 M NaCl, 8 M urea, 1.5 M β-ME), respectively, and centrifuged at 10,000 r/min for 10 min. The absorbance value of the supernatant was measured, and the protein concentration was calculated. The ionic bond, hydrogen bond, hydrophobic interaction, and disulfide bond contents were determined by calculating S2-S1, S3-S2, S4-S3, and S5-S4, respectively.

### 2.6. Statistical Analysis

Each group of experiments was repeated at least three times to reduce the error, and the data were analyzed by one-way ANOVA using statistical analysis software (SPSS 25.0) and Duncan’s test.

## 3. Results and Discussion

### 3.1. Gel Strength and WHC

The gel strength is an index to characterize the intermolecular aggregation ability of proteins, and Figure 1 shows the changes in the gel strength and WHC of duck MP after KSDF addition and ultrasonication treatments. Compared with the blank control group, the gel strength of the MP gel added with 1% of KSDF was significantly improved (*p* < 0.05), which was further enhanced by ultrasound treatment. With increasing ultrasonic power, the gel strength grew, peaked at 400 W, and then began to decline. Notably, the combined KSDF and ultrasound treatments increased the gel strength by 109.58%. The significant improvements in gel texture by KSDF and ultrasound treatment were compatible with the study of Zhang et al. [11], who suggested that the addition of dietary fiber combined with ultrasonic pretreatment significantly improved the quality of meat products. The association of proteins with KSDF through hydrophobic interactions to generate a dense three-dimensional network structure may be the cause of the increase in gel strength [24]. The cavitation shearing effect of ultrasound exposed the hydrophobic groups of the proteins to form a tightly structured three-dimensional network, especially at 400 W. However, excessive ultrasonic power interfered with the interaction between the protein and KSDF, altering the protein structure and resulting in a reduction in gel strength [25].

WHC is a quantitative indicator of the gel stability, with high WHC values corresponding to high amounts of water retained within the gel network and the gel has high-quality properties [26]. Protein gels can capture and hold water more effectively due to their special three-dimensional network structure, and a dense network structure generally tends to capture more water [27]. As shown in Figure 1, the addition of 1% KSDF improved the WHC values of the gel, consistent with the previous studies examining the effects of other types of dietary fiber addition. On the other hand, dietary fibers are water-absorbent and swellable, contributing to improving water retention [12]. The WHC of MP rose and then reduced as ultrasonic power increased; the group with the highest WHC value was that of 1%-400 W, which was consistent with the gel strength’s observation [28]. These results showed that ultrasonic treatment can be an effective approach to improve water retention in protein gels [21]. The improvement of WHC was due to the easier access of water molecules into dietary fiber–protein gel structure aided by the hole effects produced during ultrasonic treatment, while the subsequent decrease in WHC values was related to the excessive ultrasound power which caused protein degradation and weakened protein–water interactions [26].

### 3.2. Dynamic Rheological Analysis

The storage modulus is generally used as an indicator of the elasticity of the sample. G’ values of myofibrillar proteins from duck meat with different treatments were significantly altered by KSDF and ultrasonic treatment. Figure 2 illustrated the changes in G’ values across all samples as a function of temperature, showing that the three stages of the variation of G’ values with temperature were as follows. The variation of G’ in the temperature region of 25–45 °C was largely dependent on the changes in fluid flow characteristics. The change in G’ values from 45–53 °C was mainly due to the denaturation of myosin heavy chain and actin [15]. The second stage at 53–58 °C was the gel weakening period, which was caused by the denaturation of the myosin filament. Finally, the gel strengthening period occurred between 58–80 °C and 80–4 °C [29]. As shown in Figure 2, the gel formation period in the temperature range of 20–53 °C showed the increased G’ value in all experimental groups. The addition of KSDF and UT promoted the denaturation of myosin and actin, which led to an increase in G’ during the gel formation phase.

The addition of 1% KSDF significantly increased the G’ value (Figure 2), demonstrating that 1% KSDF can improve the rheological properties of protein gels during heat treatment, which was similar to the findings of Wang et al. [6]. The G’ of MP was additionally dramatically raised by the ultrasound therapy, and the protein gel with viscoelasticity was formed as the ultrasound treatment’s G’ value of MP rose with increasing ultrasound power. The hydrophobic groups of the protein molecules were exposed as a result of the cavitation effect of ultrasound, and the hydrophobic contacts at the surface were stronger as the ultrasound power rose [30].

### 3.3. Water State and Distribution of MP Gel

By evaluating the T2 relaxation time in the sample, low-field NMR is frequently employed as an important technique to analyze the distribution of moisture in foods [31]. The relaxation time curve for each gel sample was composed of three main peaks, among which the peak within 0–10 ms (T2b) indicated the bound water tightly associated with protein; the second peak ranged from 100 ms to 1000 ms (T21) was defined as immobilized water, which represented the intra-myofibrillar water; and the T22 (over 1000 ms) was named as free water, corresponding to extra-myofibrillar water [32].

The T21 of MP gels with KSDF and ultrasound treatment changed to a shorter relaxation time when compared to MP gels without any treatment, suggesting that the immobilized water had poor mobility and was more intimately bound to the proteins, in agreement with [33] who reported that ultrasound caused lower moisture mobility. According to Figure 3, the amount of bound water did not change dramatically with the addition of KSDF and ultrasound-assisted, but the quantity of immobilized water increased, and the amount of free water dropped. Based on the findings, there was less water loss and increased water retention because the water molecules and the hybrid gel were more tightly associated [34]. It was explained by the fact that a three-dimensional network structure that can trap more water molecules is produced as a result of an increase in surface hydrophobicity and hydrophobic interactions, which is also compatible with the findings of Wang et al. [35].

### 3.4. Microstructure Analysis

Scanning electron microscopy (SEM) was used to demonstrate the microstructure of the gel. A more stable three-dimensional network structure was produced as a result of MP sols being denatured and aggregated during heating. With the addition of KSDF, the three-dimensional network structure of myofibrillar proteins was denser and more symmetrical (Figure 4A). It was evident that the gel network became continuous and fine, thus explaining that the addition of KSDF facilitated the formation of the gel network structure and further improved the gel strength. While certain discontinuous structures started to emerge as the ultrasonic power rose, it was also clear that other water channels had filled in or even vanished. There were significantly fewer water channels, which could be one of the reasons explaining the higher gel strength [12]. According to research, tertiary and secondary protein structures can be changed by appropriate ultrasonic processing, which enhances the gel structure [36]. However, at an ultrasound power of 600 W, the gel network was scale-like and the continuous part was smoother, which may be due to the excessive ultrasound power changing the structure of myofibrillar proteins and leading to a disordered protein aggregation [33].

### 3.5. Changes the Solubility

Solubility is an indicator of the degree of protein aggregation and denaturation [37], and Figure 5A suggested that the solubility of the proteins was increased with the increase in the ultrasound power, reaching a maximum at 400 W. The solubility decreased again when the ultrasound power was increased. When 1% KSDF was added, myofibrillar proteins were also considerably more soluble than in the control group (*p* < 0.05), which may be since the KSDF’s enhanced polarity when attached to myofibrillar proteins made it simpler for the protein’s hydrophilic groups to bind to water [38]. The aggregation state between protein molecules and water was altered by the physical cavitation of ultrasound, producing smaller aggregates with a greater surface area [39], thus increasing the soluble protein content. However, at the power of 600 W, the high intensity of ultrasound may disrupt the structure of myofibrillar protein and inhibit the hydrophilic groups of the protein from binding to water, which caused a reduction in solubility.

### 3.6. Surface Hydrophobicity

Figure 5B demonstrates the reduction in hydrophobicity with the addition of 1% KSDF, which probably be brought on by the binding of dietary fiber molecules to proteins. For instance, protein aggregation may be facilitated by the binding of dietary fiber to proteins, which lowers the hydrophobicity of the protein surface [36]. The tertiary structure of the protein molecules may unravel as the sonication treatment’s power increases, which could affect hydrogen bonds, electrostatic interactions, and hydration between protein molecules. Surface hydrophobicity may also increase as a result of the exposure of buried hydrophobic groups in the protein molecules [40]. The enhanced surface hydrophobicity promotes the formation of hydrophobic aggregates, resulting in a tighter protein structure and a more stable gel structure.

### 3.7. Total SH

Sulfhydryl groups are considered to be important for protein structure and certain redox reactions in living organisms, and they also play a crucial function in maintaining the conformation of proteins. Figure 5C demonstrated that while the KSDF addition had no discernible impact on the gels’ overall sulfhydryl content, the KSDF composite protein system’s sulfhydryl content dramatically increased after being ultrasonically processed. KSDF functions as an antioxidant that destroys hydroxyl radicals, and the combination of ultrasound treatment significantly improved this scavenging ability, which mitigated the loss of sulfhydryl groups and led to an increase in the sulfhydryl content [18]. As a result, the total sulfhydryl content increased dramatically with increasing ultrasound power.

### 3.8. SDS-PAGE

The quality of meat products is greatly influenced by the gel structure created when myofibrillar proteins are heated. Myosin heavy chain, actin, and light chain are each depicted on the electrophoretic bands in Figure 6A. There were no significant changes in the electrophoretic bands of the first four groups of samples, indicating that the addition of KSDF and appropriate sonication did not change the structure of duck myofibrillar proteins, without obvious breakage and formation of covalent bonds in the protein. This observation was consistent with the results of the ultrasonic modification of egg white protein gels [36]. However, the electrophoretic bands of the KSDF–MP complex changed significantly when the sonication power was increased to 600 W. This may be since the relative high sonication power changed the subunits of the myofibrillar protein and degraded some of the duck myofibrillar proteins [41]. It is possible that the thermal effect and mechanical shear stress that occurred during the sonication process, which caused the mixed protein solution to become a gel when the sample was treated, is what caused the protein electrophoresis bands under 600 W treatment to be generally consistent with the thermally induced gel bands in Figure 6B.

### 3.9. Particle Size and Zeta Potential

Table 1 demonstrates that the addition of KSDF resulted in a significant reduction in the particle size of the MP protein, which may be due to the fact that the KSDF and protein molecules were bonded in either covalent or non-covalent manners, forming aggregates with a smaller surface area, thus resulting in a reduction in particle size. The particle size of protein molecules was also significantly reduced after ultrasound treatment. It has been hypothesized that the cavitation effect of ultrasound disrupts the non-covalent connections between MP molecules, causing the breakup of protein aggregates and a decline in particle size [23]. It is worth noting that the particle size of KSDF and MP aggregates increased significantly at an ultrasound power of 400 W. This is probably a result of the system’s KSDF forming molecular clusters with the MP in a more stable bound way during ultrasonic treatment at 400 W, which led to an increase in particle size. It has been hypothesized that ultrasound mostly disrupts non-bonded contacts, and that its impact varies with the ionic strength of the solution [42], and that the solution system is probably most stable when treated at 400 W.

The ζ-potential reflected the net charge of the protein surface and as can be seen in Table 1 the potentials were all negative, indicating that more negatively charged amino acids were in proteins than positively charged ones. A decrease in the absolute value of the ζ-potential indicated that the protein dispersion system was becoming less stable or aggregating. The absolute value of the ζ-potential decreased significantly after the ultrasonic treatment, which may be due to the local overheating that occurred during the ultrasonic process [43]. Protein molecules can be unfolded by ultrasound treatment, their charge distribution can be changed, positively charged amino acids can be exposed, and the negative charge on the protein surface can be neutralized. Accordingly, the reduction in net surface charge may be related to the unfolding of tertiary structures exposing hydrophobic non-polar residues [44].

### 3.10. Changes in Secondary Structure

Figure 7A reflects the effect of different treatments on the secondary structure of duck myofibrillar protein. The amide I band is 1600–1700 cm^−1^, which reflects the vibrational stretching of the carbon–oxygen double bond and has high sensitivity and is generally used as an indicator of the secondary structure of the protein, including α-helix, β-sheet, β-turn, and random coil. Analysis of the IR profiles of the different treatments on the duck myofibrillar proteins showed no significant change in the characteristic peaks for each group of samples, indicating that the addition of KSDF or ultrasound treatment did not change the type of functional groups. In the experimental group, the addition of KSDF shifted the location of the maximum absorption (1000–1200 cm^−1^), this change may have been brought on by the addition of dietary fiber, in line with the findings of the study of Kobayashi et al. [45].

In accordance with Figure 7B, the contents of β-sheets and β-turns did not change considerably, the contents of random coils reduced with the addition of dietary fiber, and the contents of α-helices increased with the rise in ultrasonic power. The results suggested that the random coil may be transformed into other structures with higher stability, similar to the research of Du et al. [17]. The addition of KSDF and ultrasound treatment with 400 W dramatically raised the proportion of α-helices in the experimental group to which KSDF was added, probably due to the formation of more α-helices structures and its higher stability, as observed by Yu et al. [46].

### 3.11. Chemical Forces Analysis

The interactions of the myofibrillar proteins after the addition of KSDF and sonication treatment are shown in Figure 8. It demonstrated that the ionic, hydrogen, and disulfide bonds of the system were significantly reduced with the addition of KSDF, with no significant difference in hydrophobic interactions, but protein hydrophobic interactions significantly increased after sonication. The lower proportions of ionic and hydrogen bonds in the composite system of KSDF and myofibrillar protein suggested that the chemical forces maintaining the structural stability of the gel network may not be mainly dependent on ionic or hydrogen bonds. The hydrophobic interactions increased with increasing ultrasound power, this is most likely as a result of the protein’s molecular structure’s hydrophobic groups being continuously affected by ultrasonic impacts, which cause tertiary structures to unfold, resulting in increased hydrophobic interaction forces. The disulfide bonding content was significantly lower with the addition of KSDF only compared to the control, attributed to the non-disulfide bonding polymerization reaction between the KSDF and MP molecules by binding some of the thiol groups during the heating process [23]. However, after ultrasound treatment, the disulfide bonds of the system increased significantly, probably because the cavitation of ultrasound generated free radicals such as H- and -OH. Therefore, the increasing number of sulfhydryl groups further formed disulfide bonds under the oxidation of these free radicals, causing the already loose protein structure to become rigid, a result which was also in line with the gel strength [19].

## 4. Conclusions

The addition of KSDF alone was able to somewhat increase the MP gel’s gel strength and WHC while also causing a more even distribution of the MP gel’s microstructure. In particular, it was demonstrated that the quality of MP gels was significantly improved by the synergistic interaction between KSDF and ultrasonication. The gel strength, water-holding capacity, rheological characteristics, solubility, and hydrophobicity of MP gels treated with ultrasonication and KSDF were all better than when KSDF was added alone. Particularly during 400 W ultrasonication, KSDF and duck myofibrillar proteins were more closely bound by ultrasonication, causing a shift in the secondary structure of KSDF–MP from disordered to ordered α-helices, which resulted in the creation of a more stable three-dimensional network structure of MP gels. Ultrasound-assisted KSDF treatment improved the meat quality of duck breast by modifying the structure and gel properties of MP. With the help of this study, duck meat’s myofibrillar protein gel qualities can be enhanced, and the groundwork is laid for the use of ultrasound as a green technology in the creation of meat products with functional properties.

## Figures and Tables

**Figure 1 foods-11-03998-f001:**
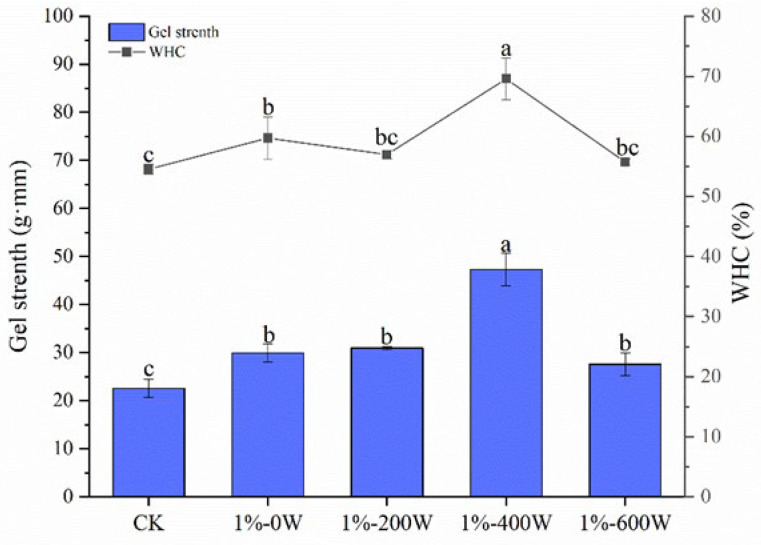
Effects of different treatments on MP gel strength and WHC, the bands from left to right are the control group, 1% KSDF–MP treated with ultrasonic energy at 0 W, 200 W, 400 W, and 600 W, respectively, different letters in the same indicator indicate significant differences (*p* < 0.05).

**Figure 2 foods-11-03998-f002:**
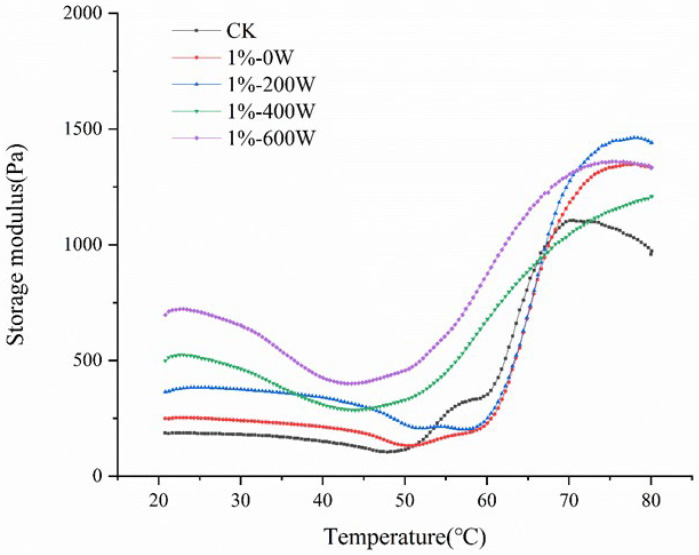
Storage modulus (G’) of MP gel with different treatments.

**Figure 3 foods-11-03998-f003:**
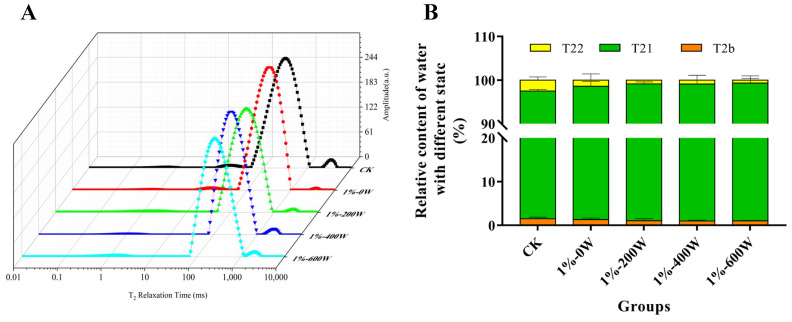
Effects of different groups on the water relaxation time (**A**) and relative content of water with different states of composite gels (**B**).

**Figure 4 foods-11-03998-f004:**
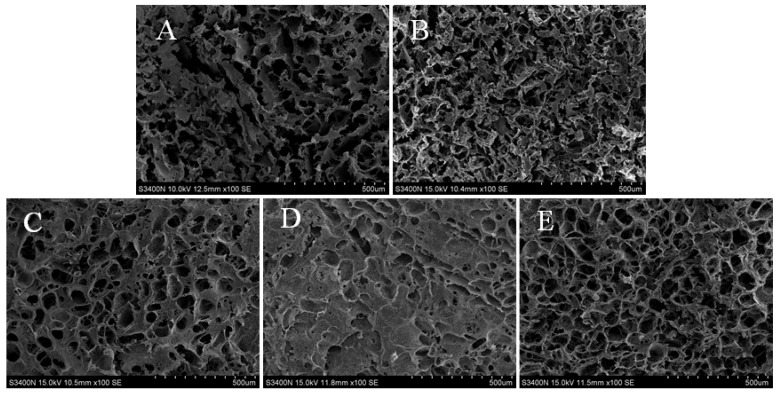
SEM photographs of the different treatment groups on MP gel, (**A**–**E**) indicate the gel with CK, 1%, 1%-0 W, 1%-200 W, 1%-400 W, and 1%-600 W, respectively.

**Figure 5 foods-11-03998-f005:**
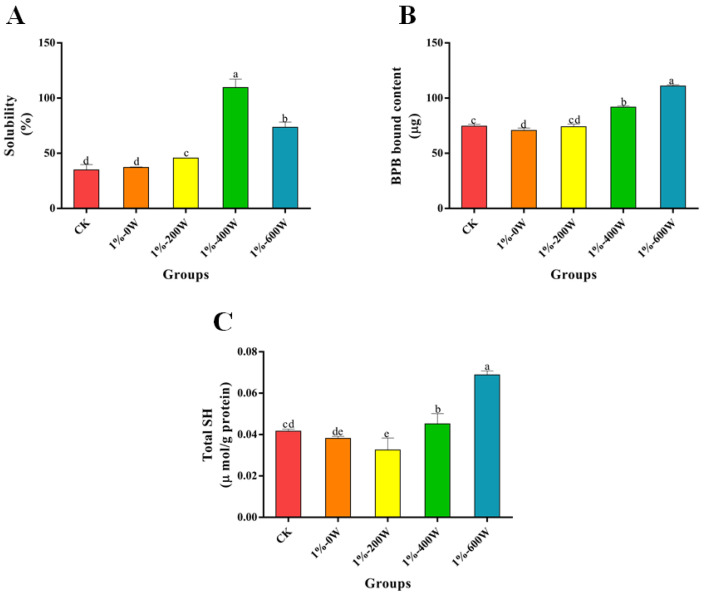
Effect of different treatments on the solubility (**A**), surface hydrophobicity (**B**), and total sulfhydryl groups (**C**). Different letters at solubility, BPB bound content and Total SH differ significantly (*p* < 0.05).

**Figure 6 foods-11-03998-f006:**
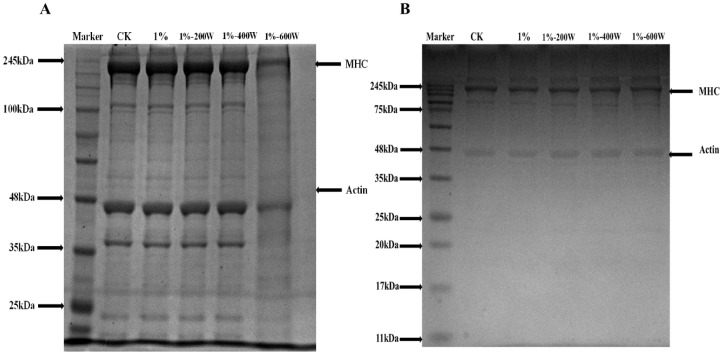
SDS-PAGE of MP samples with different treatments (**A**), the MP gels after different treatments (**B**).

**Figure 7 foods-11-03998-f007:**
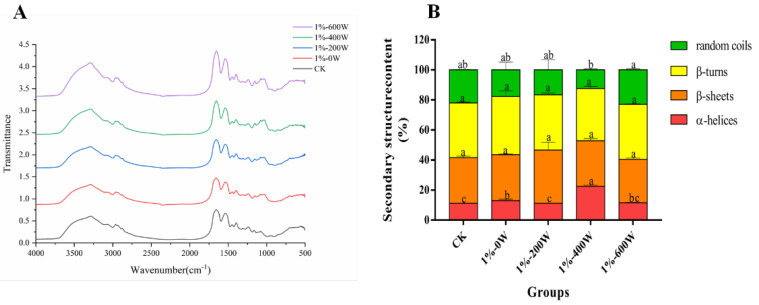
Effect of different treatments on the FTIR (**A**) and secondary structure (**B**) of MP. Different letters in the same color group represent significant differences (*p* < 0.05).

**Figure 8 foods-11-03998-f008:**
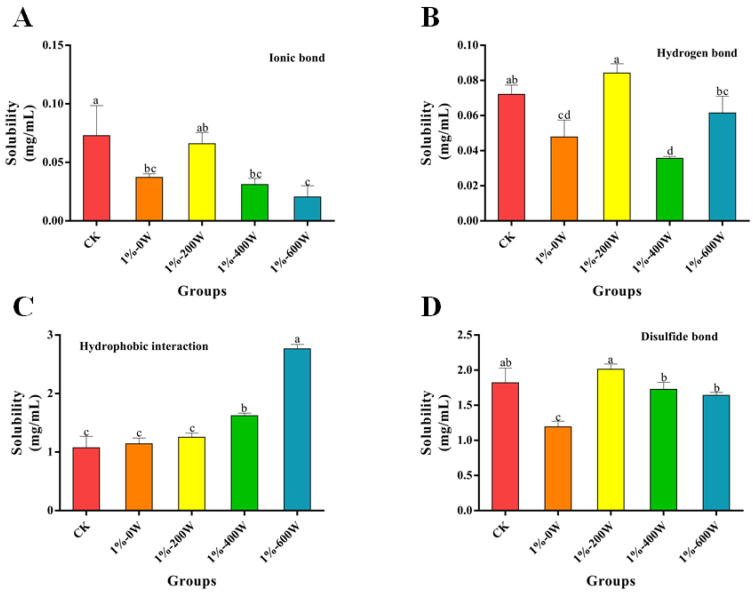
Effect of different treatments on MP chemical forces. A-D indicate the content of ionic bond, hydrogen bond, hydrophobic interaction and disulfide bond for different groups of samples, respectively. Different letters indicate significant differences between the chemical forces of samples from different groups (*p* < 0.05).

**Table 1 foods-11-03998-t001:** Effect of different treatments on the particle size and zeta potential of MP.

Groups	Size (μm)	Zeta Potential (mV)
CK	27.40 ± 0.37 b	−25.73 ± 1.60 c
1%-0 W	20.15 ± 0.10 c	−20.43 ± 0.59 b
1%-200 W	19.47 ± 0.25 c	−18.00 ± 0.78 a
1%-400 W	31.52 ± 0.44 a	−16.83 ± 1.10 a
1%-600 W	15.62 ± 0.76 d	−18.47 ± 0.42 a

Results are presented as the mean ± standard deviation. Different letters in the same column represent significant differences (*p* < 0.05).

## Data Availability

The data is included in the article.

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
