# Peer review of "Interaction between Kidney-Bean Polysaccharides and Duck Myofibrillar Protein as Affected by Ultrasonication: Effects on Gel Properties and Structure"

_foods, 2022, doi:10.3390/foods11243998_

Round 1

Reviewer 1 Report

Line 24 - abbreviate myofibrillar protein

Affluenza has no scientific meaning or application

English language should be extensively reviewed

The lack of text cohesion and coherence hinder the comprehension of the introduction. 

The methodology is coherent with the proposed objectives. They are well presented. All units should be reviewed

Figures' title should be self-explanatory

In my opinion, to work with animal´s meat it is needed that the research is approved by an ethics comittee.

Author Response

Thank you for the comments. We have revised the manuscript according to the suggestions.The related changes have been made according to your comments in the revised manuscript carefully.A list of changes is summarized in the letter , point by point. Simultaneously, all the corrections have been highlighted using red color in the revised manuscript. We hope these modifications meet the requirement.

Point 1: Line 24 - abbreviate myofibrillar protein

Response 1: Thank you for such professional advice. I have read the sentence carefully and found that the expression was indeed inappropriate, and we have revised the sentence. You can see the revised content in line 24 of the article.

Point 2: Affluenza has no scientific meaning or application

Response 2: Thank you for your revisions. I have read the sentence carefully and found that the expression was indeed inappropriate, and we have revised the sentence. You can see the revised content in lines 34-36 of the article. “There is a tendency toward creating various functional low-temperature minced meat products due to the gradual rise in the number of sub-healthy people”

Point 3: English language should be extensively reviewed

Response 3: Thank you for your advice, I have read the whole article again carefully and reviewed the English language. Changes to the manuscript have been made in red in the original text.

Point 4: The lack of text cohesion and coherence hinder the comprehension of the introduction

Response 4: Thank you for your suggestions, this is helpful. For the introduction section, I reorganized the logic and made the following changes, including adding some background, please see the reddened section of my re-uploaded manuscript.

Point 5: Figures' title should be self-explanatory

Response 5: Thank you for your suggestion, this is a point that we did not explain clearly, thank you very much for the point. For ease of understanding, for the figures' title I have added the following” the bands from left to right are the control group, 1% KSDF-MP treated with ultrasonic en-ergy at 0W, 200W, 400W, and 600W, respectively.”

Point 6: In my opinion, to work with animal´s meat it is needed that the research is approved by an ethics comittee.

Response 6: Thank you for your comment. Research procedures must be carried out in accordance with national and institutional regulations. Statements on animal welfare should confirm that the study complied with all relevant legislation. It is generally accepted that ethical committee approval is required when clinical trials involving animals are involved. In this manuscript we did not perform animal experiments and sensory evaluations, therefore an ethical proof is not required. On the other hand, the experimental material in our study, duck breast, was purchased from the market and is edible meat that can be circulated in the market. Thank you again for your advice!

Reviewer 2 Report

The authors explored the effect of ultrasound on the interaction between kidney bean dietary fiber (KSDF) and duck myofibrillar proteins (MP). In addition, they investigated the link between the physicochemical properties of KSDF-proteins and the properties of gels. The study was well designed and good results were obtained.

I have only one recommendation. I would like the passage to be rewritten because I am in doubt about the information, as shown in figure 5A..."The solubility of myofibrillar protein was also significantly higher in the 1% KSDF addition than in the control group (p < 0.05), which may be due to the increased polarity of the KSDF when combined with myofibrillar protein, and the hydrophilic groups of the protein binding to water thus enhancing the interaction force between protein and water".

Author Response

Thank you for the comments. We have revised the manuscript according to the suggestions.The related changes have been made according to your comments in the revised manuscript carefully.A list of changes is summarized in the letter , point by point. Simultaneously, all the corrections have been highlighted using red color in the revised manuscript. We hope these modifications meet the requirement.

Point 1: I have only one recommendation. I would like the passage to be rewritten because I am in doubt about the information, as shown in figure 5A..."The solubility of myofibrillar protein was also significantly higher in the 1% KSDF addition than in the control group (p < 0.05), which may be due to the increased polarity of the KSDF when combined with myofibrillar protein, and the hydrophilic groups of the protein binding to water thus enhancing the interaction force between protein and water".

Response 1: Thank you for the comments. We have carefully revised the introduction according to the suggestions. See it in lines 284-287.” When 1% KSDF was added, myofibrillar proteins were also considerably more soluble than in the control group (p < 0.05), which may be since the KSDF's enhanced polarity when attached to myofibrillar proteins made it simpler for the protein's hydrophilic groups to bind to water [38]”

Reviewer 3 Report

The work was well founded and the experimental design robust. The discussion and interpretation of the results is detailed, pertinent, and clearly relates all the response variables, explaining in detail the adequate effect of the addition of fiber and the relevance of the use of ultrasound, advancing pertinent explanations about the specific action of these on the modification of protein structure and desired induction of possible protein-fiber interactions. Despite the good results and the relevance of the discussion of results, the authors do not specify the contributions that their results have made to the use of ultrasound in gel technology. It is recommended that they add a small paragraph in which they point out the contributions in a timely manner. of their results to the knowledge, particularly of the effect of ultrasound on the structuring of gels.

Author Response

Thank you for the comments. We have revised the manuscript according to the suggestions.The related changes have been made according to your comments in the revised manuscript carefully.A list of changes is summarized in the letter , point by point. Simultaneously, all the corrections have been highlighted using red color in the revised manuscript. We hope these modifications meet the requirement.

Point 1: The work was well founded and the experimental design robust. The discussion and interpretation of the results is detailed, pertinent, and clearly relates all the response variables, explaining in detail the adequate effect of the addition of fiber and the relevance of the use of ultrasound, advancing pertinent explanations about the specific action of these on the modification of protein structure and desired induction of possible protein-fiber interactions. Despite the good results and the relevance of the discussion of results, the authors do not specify the contributions that their results have made to the use of ultrasound in gel technology. It is recommended that they add a small paragraph in which they point out the contributions in a timely manner. of their results to the knowledge, particularly of the effect of ultrasound on the structuring of gels.

Response 1: Thanks very much for your professional suggestion, it has helped us a lot. We point out the effect of ultrasound on the gel structure in the exposition of each experimental result. In addition to this, you can see our changes in lines 417-424 of the article.” Particularly during 400W ultrasonication, KSDF and duck myofibrillar proteins were more closely bound by ultrasonication, causing a shift in the secondary structure of KSDF-MP from disordered to ordered α-helices, which resulted in the creation of a more stable three-dimensional network structure of MP gels. Ultrasound-assisted KSDF treatment improved the meat quality of duck breast by modifying the structure and gel properties of MP. With the help of this study, duck meat's myofibrillar protein gel qualities can be enhanced, and the groundwork is laid for the use of ultrasound as a green technology in the creation of meat products with functional properties.”
